# Physicochemical Properties of 3D-Printed Polylactic Acid/Hydroxyapatite Scaffolds

**DOI:** 10.3390/polym15132849

**Published:** 2023-06-28

**Authors:** Sara Pérez-Davila, Natalia Garrido-Gulías, Laura González-Rodríguez, Miriam López-Álvarez, Julia Serra, José Eugenio López-Periago, Pío González

**Affiliations:** 1CINTECX, Universidade de Vigo, Grupo de Novos Materiais, 36310 Vigo, Spain; nataliagarridogulias@gmail.com (N.G.-G.); laugonzalez@uvigo.gal (L.G.-R.); miriammsd@uvigo.gal (M.L.-Á.); jserra@uvigo.gal (J.S.); pglez@uvigo.gal (P.G.); 2Galicia Sur Health Research Institute (IIS Galicia Sur), SERGAS-UVIGO, 36213 Vigo, Spain; 3Área de Edafoloxía e Química Agrígola, Departamento Bioloxía Vexetal e Ciencia do Solo, Facultade de Ciencias, Universidade de Vigo, 32004 Ourense, Spain; edelperi@uvigo.gal

**Keywords:** polylactic acid (PLA), hydroxyapatite (HA), 3D printing, physicochemical analysis, biocompatibility

## Abstract

The reconstruction or regeneration of damaged bone tissue is one of the challenges of orthopedic surgery and tissue engineering. Among all strategies investigated, additive manufacturing by fused deposition modeling (3D-FDM printing) opens the possibility to obtain patient-specific scaffolds with controlled architectures. The present work evaluates in depth 3D direct printing, avoiding the need for a pre-fabricated filament, to obtain bone-related scaffolds from direct mixtures of polylactic acid (PLA) and hydroxyapatite (HA). For it, a systematic physicochemical characterization (SEM-EDS, FT-Raman, XRD, micro-CT and nanoindentation) was performed, using different PLA/HA ratios and percentages of infill. Results prove the versatility of this methodology with an efficient HA incorporation in the 3D-printed scaffolds up to 13 wt.% of the total mass and a uniform distribution of the HA particles in the scaffold at the macro level, both longitudinal and cross sections. Moreover, an exponential distribution of the HA particles from the surface toward the interior of the biocomposite cord (micro level), within the first 80 µm (10% of the entire cord diameter), is also confirmed, providing the scaffold with surface roughness and higher bioavailability. In relation to the pores, they can range in size from 250 to 850 µm and can represent a percentage, in relation to the total volume of the scaffold, from 24% up to 76%. The mechanical properties indicate an increase in Young’s modulus with the HA content of up to ~50%, compared to the scaffolds without HA. Finally, the in vitro evaluation confirms MG63 cell proliferation on the 3D-printed PLA/HA scaffolds after up to 21 days of incubation.

## 1. Introduction

The tissue regeneration of massive bone defects caused by trauma, infection, tumor resection and congenital disabilities is a major treatment challenge in plastic, reconstructive and orthopedic surgery [1]. Currently, autologous bone grafts are considered the gold standard for treating such defects and promoting bone regeneration. They offer better functional responses compared to synthetic materials and eliminate the risk of graft rejection. However, they present limitations in certain types of surgery due to their high levels of resorption, low availability in the case of high volumes required and donor-site morbidity [2,3]. These shortcomings require the research of new bone grafts and scaffolds where the chemical composition, porosity (size, volume, interconnectivity) and mechanical strength are critical parameters that will define their performance [4]. Ideally, a bone scaffold should fulfill multiple functions, including facilitating cell delivery, promoting the differentiation of regenerative cells and supporting new bone growth. Therefore, it should be biocompatible, osteoconductive, osteoinductive and biodegradable at a controlled rate [1]. Additionally, the ability to manufacture the scaffold in irregular shapes is also essential.

Several methods have been proposed in the literature [5] for manufacturing porous bone scaffolds, including chemical/gas foaming, solvent casting, particle/salt leaching, freeze drying, thermally induced phase separation, foam gel, electrospinning and stereolithography. However, these methods have limitations in fully controlling pore size, interconnectivity and tailored scaffolds for specific defects and shapes. Additive manufacturing by 3D-FDM (fused deposition modeling), recently incorporated in the biomedical field, offers the required versatility for the design and fabrication of scaffolds with the specific requirements of porosity and shape [4]. This 3D-printing methodology, when combined with the right materials and optimized printing parameters, allows for the fabrication of patient-specific implants/scaffolds with precise architectural control. This is achieved by using an STL file format obtained from clinical images of the patient, taking us one step closer to personalized medicine [1,6,7,8].

In relation to the materials, polylactic acid (PLA) is highlighted as a bioabsorbable, biodegradable and biocompatible polymer, representing a promising alternative to traditional biomaterials and non-biodegradable polymers [9]. The use of biodegradable polymer-based scaffolds offers the advantage of gradual replacement by new tissue during the degradation process, eliminating the need for additional interventions to remove the biomaterial from the body [10]. In fact, this natural polymer can be degraded by enzymatic activity or hydrolysis, forming lactic acid which is usually present in the body, preventing inflammatory reactions. Due to these properties, PLA is widely employed in almost all medical specialties, including orthopedic applications [11,12,13]. There is, finally, a property of PLA of particular interest in relation to the 3D-FDM technology: its low glass transition temperature (55–65 °C) [14]. This property makes it deformable under high temperatures (190–220 °C), providing the versatility sought in the manufacture of the scaffold [11,15,16,17].

In addition to porosity and composition, the mechanical strength of the scaffold is, as previously mentioned, another critical parameter to be considered. It is well-known that the low mechanical properties provided by PLA would not allow maintaining the morphology of the scaffold under the load of the musculoskeletal system [10]. Therefore, in order to satisfy this requirement, several strategies have been proposed, i.e., the combination of PLA with other materials in a biocomposite of them. In this line, current research in resorbable orthopedic implants has focused on combining degradable polymers with bioactive ceramics. Specifically, the addition of small-volume fractions of hydroxyapatite (HA) to PLA has been proven to enhance the scaffold’s strength by improving stiffness and partially counteracting the pH effects [18]. Moreover, the incorporation of HA guarantees a higher similarity of the biocomposite to the natural bone tissue. Its presence in the polymeric matrix would then provide it with the desired osteoconductive, osteoinductive [19,20,21] and osteogenic properties [2].

In relation to 3D-printed PLA, numerous studies have been found in the literature reviewed where PLA has been combined with HA by the surface modification of 3D-printed parts [4,22] or by manufacturing a filament prior to the printing process [16,23,24,25]. However, obtaining this single pre-fabricated filament requires a laborious process, especially when the materials to combine are formulated in different sizes. For instance, in the case of a composite filament using PLA pellets and HA powder, the pellets are subjected to a freezing process at −80 °C for a certain time to make them brittle and then ground and sifted to get smaller sizes closer to the HA powder. Following this process, a filament with a uniform distribution of HA particles inside the PLA matrix is obtained [26]. In addition to this laborious process, the methodology based on pre-fabricated filaments causes a loss of versatility in the 3D-printing process, hindering the obtaining of personalized 3D-printed scaffolds with a composition ad hoc.

The direct combination of PLA and HA pellets in the desired rate to instantaneously obtain the personalized 3D-printed scaffold would offer versatility to the process by easily modulating the PLA/HA ratio. To address this, the present study proposes directly fabricating 3D-printed PLA/HA scaffolds with modulated composition and porosity properties by a one-step integration approach from the combination of the two materials in different formulations. Moreover, a systematic physicochemical characterization is provided to evaluate their composition, microstructure and mechanical properties (SEM-EDS, FT-Raman, XRD, micro-CT and nanoindentation). Furthermore, a preliminary in vitro evaluation is performed to validate the expected viability of the cell line MG63 when seeded in the 3D-printed PLA/HA scaffolds for up to 21 days of incubation. The present work aims to validate the 3D-printing methodology proposed and the modulated scaffolds obtained, in different PLA/HA ratios and porosity ranges, for bone tissue regeneration applications.

## 2. Materials and Methods

### 2.1. Starting Materials and Scaffold Printing

Pellets, with dimensions of 5 × 3.5 mm in oval shape, of natural polylactic acid (PLA) SMARTFIL^®^ were purchased from Smart Materials, Jaén, Spain. The main properties are summarized in Table 1. As hydroxyapatite (HA), we used hydroxylapatite Captal^®^ ‘R’ (batch P120R) powder with spherical morphology (average particle size 3.29 µm), acquired from Plasma-Biotal Limited (Tideswell, UK). According to the manufacturers, it presents a Ca:P ratio in the range of 1.66–1.72, a crystallinity of around 85–95% and a high surface area of typically 6–20 m^2^/g (http://www.plasma-biotal.com/captal-r-hydroxylapatite/, accessed on 2 March 2023).

The desired amount of PLA was carefully and repeatedly mixed with that of HA in a Petri dish with a spatula to unite them into a single mass; the total mass prepared was 10 g. The PLA and HA fractions incorporated in the mixtures and their respective infill percentages defined for printing are summarized in Table 2. Once obtained, the single mass was next introduced in the 3D-FDM printer hopper (TUMAKER Voladora NX Pellet, Tumaker SL, Oiartzun, España) to obtain the 3D-printed PLA/HA scaffolds shaped as discs. Two sets of PLA/HA scaffolds were first designed using SolidWorks 2016 software as follows: (1) discs with dimensions of 12 mm diameter and 3 mm height with variation in the infill percentage from 60 to 100 to be used for the physicochemical characterization; and (2) discs of 8 mm diameter and 2.5 mm height for the biological tests. The digital data from the designs were then saved as STL files to generate the corresponding G-code sets for 3D printing through the Simplify3D Professional Software version 4.1.2 (Figure 1). The 3D-FDM printer has two-point temperature control for each of the extruders, and it was first adjusted to T1 = 140 °C and T2 = 220 °C, respectively. However, the latter (T2) was modified to a range from 220 to 240°C to favor the printing of scaffolds with the highest wt.% in HA. The main 3D-printing parameters are summarized in Table 3.

### 2.2. Physicochemical Characterization

The global structure of the scaffolds was first analyzed with a stereomicroscope Nikon SMZ25 and the surface morphology with a scanning electron microscope (SEM) JEOL JSM-6700F high-resolution (JEOL Ltd., Tokyo, Japan). The elemental composition was detected by EDS using an Oxford Inca Energy 300 (Oxford Instruments, Oxford, UK) coupled to the SEM microscope. The crystalline structure was evaluated by X-ray diffraction (XRD) in an X’Pert Pro Panalytical diffractometer (Malvern Panalytical, Malvern, UK) with monochromated Cu-Kα radiation (λ = 1.5406 Å) and with a 2θ range of 4–100°. Moreover, FT-Raman spectroscopy was carried out to identify the main molecular vibrations and corresponding functional groups, using a B&W Tek i-Raman-785S instrument (Metrohm, Herisau, Switzerland) equipped with a BAC 100 Probe (785 nm) in the wavenumber range from 250 to 3250 cm^−1^ and a maximum incident laser radiation of 340 mW. The mechanical properties were analyzed using a nanoindenter XP (MTS Nano Instruments Inc, Oak Ridge, TN, USA) where hardness and Young’s modulus values were measured using a 100 nm radius triangular pyramid indenter tip (Berkovich-type indenter) with the CSM (continuous stiffness measurement) mode to perform dynamic measurements as a function of depth and XP head. A large number of indentations (30) were programmed, and the average of the valid results was calculated ± standard deviation. Finally, micro-computed tomography (micro-CT) was used to evaluate structural parameters like porosity, connectivity factor (Euler characteristic), particle size and distribution into the scaffolds, depending on wt.% incorporated and % of infill used. Micro-CT scans were acquired with YXLON FF20 CT equipment (YXLON Comet Technologies Inc, Hudson, OH, USA) using 100 kV and 35 µA. The images were taken in 720 steps with a full sample rotation, using 50 ms acquisition time for each image. These scans were reconstructed using the CERA 1.5.5.0 software (Siemens-HealthcareGmbH) to obtain the 3D models. The image analysis and evaluation were performed in Avizo3D (ThermoScientificTM AvizoTM Software 9). Image processing consisted of four main steps: (1) shifting and aligning the filling grid in orthogonal coordinates, (2) supervised image binarization to separate the fill from the voids, (3) denoising the binary image to remove small spots and (4) selecting the volume of interest that contains the fill pattern.

### 2.3. Biological Response In Vitro: Cell Proliferation

Before the cell assays, the set of 3D-printed PLA/HA discs manufactured for the biological tests was packed in a laminar flow cabin and sterilized with a dose level of between 25 and 35 kGy of gamma radiation, performed with Aragogamma S.L. (Barcelona, Spain) using a 60Co source irradiator at room temperature, as per ISO 13485:2018. The 3D-printed PLA/HA scaffolds were first placed in a 48-well microplate and then covered with a cell suspension of the human osteosarcoma cell line MG63 (ECACC, UK) of 7 × 10^4^ cells/mL in 300 µL of EMEM medium (Lonza, Basilea, Switzerland), supplemented with 10% of fetal bovine serum (Hyclone Laboratories LLC, Logan, UT, USA) and 1% of a combination of penicillin, streptomycin and amphotericin B (Lonza, Basilea, Switzerland). Empty tissue culture polystyrene (TCP) microplate wells were also seeded with the same cell suspension to be used as the gold standard to confirm the healthy stage of cells. The cells were cultured for up to 21 days at 37 °C and 5% CO_2_ in a humidified atmosphere. The culture medium was renewed every 2–3 days. Cell proliferation was quantified after 7, 14 and 21 days with the MTS Cell Proliferation Assay Kit (Abcam, Cambridge, UK). This colorimetric assay is based on the reduction of the MTS tetrazolium compound only by viable cells to generate a colored formazan dye that is soluble in the culture medium. A volume of 10 μL of MTS reactive was added to each well. After 45 min of incubation (37 °C and 5% CO_2_), the absorbance of the resulting solutions was read at a wavelength of 490 nm in a microplate spectrophotometer (Bio-Rad, Hercules, CA, USA). Five replicates per material per condition were evaluated, and the results were expressed in the percentage as mean ± standard error. Two independent experiments were performed.

### 2.4. Statistical Analysis

Biological data were analyzed using GraphPad Prism 8 (GraphPad Software Inc., San Diego, CA, USA), and the results were represented graphically as the mean ± standard error of means (mean ± SEM). The nonparametric Mann–Whitney U test was used to determine the statistical differences in the nano-indentation measurements obtained for the different PLA/HA scaffolds evaluated. The same nonparametric method was used to evaluate statistically significant differences in the biological data. Statistical significance was determined to be * (*p* ≤ 0.05) at the 95% confidence level.

## 3. Results and Discussion

### 3.1. Morphological and Elemental Composition

After the 3D-printed PLA/HA scaffolds were fabricated with different ratios of PLA/HA pellets and infill percentages, the morphology was first analyzed by stereomicroscopy. Figure 2 shows magnified stereomicroscope images of PLA0HA scaffolds with 100% of infill (a), PLA3HA-100 (b), PLA9HA-100 (c) and PLA13HA-100 (d). First of all, when observing the PLA0HA scaffolds with 100% of infill (a), it is noted that despite the requirement of 0% porosity, the resulting scaffolds are not compact, presenting uniformly distributed square-shaped pores of around 440 µm from wall to front wall. This limitation to obtaining compact scaffolds with the methodology and conditions used is directly related to the need for printing small scaffolds (discs of 12 or 8 mm diameter and 3 or 2.5 mm height) with a large-diameter nozzle (0.8 mm). These processing conditions ensure the direct obtaining of a homogeneous mixture from pellets of polymer and bioceramic powder and prevent the mixture from clogging. It is observed, however, that despite the fact that at all the PLA/HA ratios obtained in the 100% infill condition pores, as seen in the images for PLA0HA (a), PLA3HA (b), PLA9HA (c) and PLA13HA (d), the dimensions of these pores decrease as the contribution of HA in the mixture increases, with a pore size of around 380 µm in PLA3HA-100 and 300 µm in PLA9HA-100 with PLA13HA-100% being the smallest with a size of around 250 µm from wall to front wall. In relation to this, it is important to notice that while HA was gradually incorporated into the mixture (3, 9 and 13 wt.%), the printing temperature had to be increased, in the indicated range (220–240 °C), together with the printing speed and the flow of the mixture. This increased temperature and flow of material causes the printing line with high amounts of HA to expand and explains the fact that the pore size is higher in the PLA0HA-100% scaffolds in comparison to PLA13HA-100%. To confirm this, the thickness of the PLA/HA printing line of the 3D-printed scaffolds presented in Figure 2 was measured, being around 534.66 µm in PLA3HA-100 (b), 762.22 µm in PLA9HA-100 (c) and 812.48 µm in PLA13HA-100 (d). The rectilinear pattern selected (angle 45/−45°) is clearly observed in all the images (a–d) which generates an open porosity. Moreover, it is possible to observe a translucent appearance typical of PLA in the free-of-HA scaffold, PLA0HA (a), which becomes whiter when the bioceramic is incorporated.

Going into detail with the morphology and elemental composition, a SEM/EDS analysis of the 3D-printed PLA/HA scaffolds was performed. In Figure 3, micrographs and spectra obtained for the PLA0HA-80% infill (Figure 3a,b) and PLA13HA-100% infill (Figure 3c,d) are shown. When observing the pore sizes, the highest infill density sample, PLA13HA-100% (Figure 3c), presented pores of about 250 µm from one wall to the front wall, which were 800–850 µm in the lowest-infill-density sample fabricated, PLA13HA-60% (Figure 2f). In the case of PLA free of HA, the scaffolds of PLA0HA-100% (Figure 2a) presented pores of about 350 µm, being around 500 µm in the case of PLA0HA-80% (Figure 3a). The wide range of pore sizes ≥250 µm obtained is in accordance with the ranges considered to be optimal for bone regeneration. Thus, the preference of osteoblasts for pores larger than 100 µm is generally stated [4,27,28,29]. Moreover, significant bone formation has been published for 800 µm scaffolds [30] and also 250 and 400 µm ones [31], while enhanced bone formation and vascularization are reported for scaffolds with pore sizes larger than 300 µm [4,32].

Therefore, considering only these preliminary data of pore sizes on average, the most favorable 3D-printed PLA/HA scaffolds for bone regeneration would be those with less than 100% of infill. In fact, it has been published that poly (D, L-lactic acid) (PDLLA) scaffolds with pore sizes of 325 and 420 µm resulted in a well-organized type I collagen network, while a smaller pore size of 275 µm prevented osteosarcoma-derived human osteoblasts from proliferating, differentiating and producing a functional bone ECM [33]. Going back again to Figure 3, it is also noted that the incorporation of HA particles (Figure 3c) produces a rougher surface than the surfaces of the scaffolds free of HA (Figure 3a), this also being of interest for bone regeneration. Finally, in relation to the composition, the EDS spectrum for PLA13HA-100% (Figure 3b) presents the characteristic bands of HA (P and Ca), besides the C and O typical of PLA (see the spectrum of PLA0HA-80% infill (Figure 3d)). Therefore, the HA incorporated by direct 3D printing into the composite can be detected using EDS. Furthermore, from the EDS spectra, the Ca/P atomic ratio was measured, and a value of 1.71 was obtained. This value agreed with the elemental stoichiometry between 1.66 and 1.72 provided by the manufacturer and other authors [23].

### 3.2. Structural and Mechanical Analysis

Once the morphological and elemental composition was evaluated, the structural analysis of the 3D-printed PLA/HA scaffolds was also performed by FT-Raman spectroscopy and X-ray diffraction (XRD). The mechanical properties were also evaluated by nanoindentation.

The evaluation of the main molecular vibrations by FT-Raman spectroscopy and their correspondence with certain functional groups that constitute the 3D-printed scaffolds was first assessed. Figure 4a shows the Raman spectra for PLA alone and the PLA/HA scaffolds fabricated with 100% of infill. The characteristic spectrum of 3D-printed PLA [34] was clearly observed when the PLA0HA-100% scaffold was analyzed, with the corresponding bands present at 298 cm^−^^1^ and 397 cm^−^^1^, respectively assigned to bending C-O-C and C-CO groups, an intense and sharp band at 872 cm^−^^1^ attributed to C-COO stretching, at 1041 cm^−^^1^ to C-CH_3_ skeletal stretching, at 1124 cm^−^^1^ to CH_3_ asymmetric rocking, at 1455 cm^−^^1^ to CH_3_ symmetric bending, at 1770 cm^−^^1^ to C=O asymmetric stretching and, finally, at 2946 cm^−^^1^ to CH_3_ symmetric stretching. When the Raman spectra obtained for PLA3HA, PLA9HA and PLA13HA scaffolds were observed (Figure 4a), it was noted that they presented the same PLA characteristic bands, together with, as expected, the band at 962 cm^−^^1^ attributed to the PO_4_^−^^3^ symmetric stretching mode of calcium phosphates [35], which corresponds to hydroxyapatite [36]. Figure 4b presents the quantitative evaluation of the HA incorporated into the 3D-printed PLA/HA scaffolds calculated from the FT-Raman spectra (Figure 4a) by using the ratio of the HA-related band intensity and the one of the intense and sharp PLA-related band at 872 cm^−^^1^ (I962/I872). Thus, as it can be observed, the I962/ I872 ratio of the scaffolds shown in Figure 4b increases, as expected, with the wt.% of HA incorporated into them. These quantitative results easily obtained from the FT-Raman spectra can be of interest in the development of quality control techniques for 3D-printed PLA/HA scaffolds.

With the FT-Raman evaluation, the controlled incorporation of HA into PLA was then proven together with the fact that both starting materials maintain their properties, in terms of molecular vibrations, after being submitted to the 3D-printing procedure proposed. The crystalline structure of the 3D-printed PLA/HA scaffolds was next evaluated by X-ray diffraction (XRD), and the results can be seen in Figure 5. Again, PLA alone and PLA/HA scaffolds fabricated with 100% infill were the ones analyzed, and the main diffraction peaks were identified. Figure 5a presents the XRD patterns obtained for PLA alone (PLA0HA) and the PLA/HA scaffolds fabricated with the 100% infill.

The diffraction pattern obtained for PLA0HA presented a wide band between 2θ angle = 10° and 25° with a broad peak at approximately 16°, which originated predominantly from the intermediate way of ordering the polymer chains between the amorphous and crystalline forms [22]. Therefore, the pattern for PLA0HA can be associated with the semi-crystalline nature of PLA, also indicated by other authors [37,38]. The same behavior was found for the rest of the PLA/HA scaffolds where it is clearly shown that the intensity of the PLA band decreases while the HA signal increases, with the lowest intensity band obtained for the diffraction pattern of the PLA13HA scaffold with a very weak peak above 16°, which corresponds to the α-form crystals (110)/(200) [23,39]. In relation to the gradual incorporation of HA into the scaffolds, the diffraction peaks obtained correspond with the crystalline structure of hydroxyapatite Ca_5_(PO_4_)_3_(OH) with sharp and intense diffraction peaks recorded, in particular at 31.77°, 32.17° and 32.91°, respectively corresponding to the crystal planes of hexagonal hydroxyapatite (211), (112) and (300). The other reflections located at 10.81°, 25.87°, 28.06°, 28.90°, 34.06°, 39.80°, 46.64°, 49.57°, 50.40° and 53.19° respectively correspond to the (001), (002), (102), (120), (202), (310), (222), (213), (321) and (004) diffraction planes, according to the previous literature [4,23,40]. As expected, the intensity of the HA-related peaks also increases with the HA content up to a maximum in the PLA13HA sample. Figure 5b presents the quantitative evaluation of the relative intensity of the maximum HA diffraction peak, the one at 31.77° corresponding to the crystal plane (211), calculated for each corresponding XRD diffraction pattern (Figure 5a) in percentage. It can be observed that the intensity of this diffraction peak increases with the HA wt.% incorporated into the scaffolds. This behavior, again, has a linear dependence which can be used as a predictive tool to know the structural composition of scaffolds with other intermediate incorporations of HA (0–13 wt.%).

Once the semicristalline degree of the 3D-printed PLA scaffolds was proven, together with the increasing degree of PLA crystallinity with the incorporation of HA, the mechanical properties of the 3D-printed PLA/HA scaffolds were also analyzed. The hardness and Young’s modulus measurements, carried out on the PLA/HA-100% infill scaffolds with pore sizes in the range from 250 to 350 µm in distance from one wall to the front wall (Figure 2), are presented in Figure 6. Thus, as it can be observed, the value of Young’s modulus (dark gray) obtained for the PLA0HA scaffolds was 4.57 ± 0.06 GPa, which is in agreement with the literature [41]. Moreover, it is shown that this elasticity modulus is affected by the incorporation of HA into the scaffold, being higher as the HA wt.% increases, which was also noted by other authors for PLA biocomposites with 5, 10 and 15 wt.% of HA, by 3D printing with previously extruded filaments [16]. This tendency is interpreted as an increase in the material’s stiffness as the bioceramic particle concentration increases. Statistical differences (*p* < 0.05) are observed for PLA3HA and PLA9HA in relation to the PLA0HA scaffold being significantly higher. The maximum value obtained for the PLA13HA scaffold was 6.74 ± 2.06, which supposes an increase of about 50% regarding the scaffold without HA (PLA0HA). These data are in good agreement with other authors, where PLA scaffolds by 3D printing with previously extruded filaments with 15 wt.% of HA obtained values of 5.6 GPa [42]. In the present work, it was interestingly found that this tendency between Young’s modulus and the wt.% HA incorporated into the scaffold presents a behavior of linear dependence, where the slope of the equation would be equivalent to a ratio of 0.18 (GPa/wt.%HA), indicating a rate of increase of Young’s modulus with the % HA added to the biocomposite of 18%.

In relation to the hardness values (light gray), the PLA0HA scaffolds presented a value of 0.234 ± 0.003 GPa, which was again in good agreement with other authors by the injection of molded PLA [41]. For almost all the remaining samples, the trend was also the same as for the elasticity modulus, obtaining the highest hardness mean value again for the PLA13HA scaffolds. This value presents, however, a great variability for both Young’s modulus and hardness measurements. This variability comes from the surface heterogeneity of these PLA13HA scaffolds, already shown in the SEM micrograph in Figure 3c, with aggregates that cause a much lower number of valid indentations, resulting in a much larger standard error.

### 3.3. Particle Size, Distribution, Porosity and Connectivity

To evaluate in depth the distribution of the HA particles in the 3D-printed scaffold fabricate by the proposed one-step methodology, a detailed study was performed using computed tomography. In this aspect, it is important to notice that the distribution of HA in the PLA matrix will influence not only the mechanical properties but also the regeneration process given the influence of surface roughness and HA direct availability on bone cell adhesion and activity. Thus, the size of HA particles after the printing process and their distance from the surface of the PLA matrix were evaluated. In Figure 7a, the distribution of HA particles (mainly orange, yellow and green colors) embedded in the PLA matrix (blue color) can be observed in a segmented image of the PLA13HA scaffold with 60% of infill. The different color of the HA particles indicates different distances in depth from the PLA surface to the inner part of each PLA/HA printing filament. The warmer colors (orange) correspond to the HA particles situated closer to the PLA surface. A homogeneous distribution of HA particles, both the longitudinal and cross sections of the scaffolds at the macro level, can be first easily observed together in Figure 7a with the size of the HA particles being in the range from 0 to 1 × 10^−^^4^ mm^3^. Larger sizes correspond to particle aggregates caused by the stop/start-up operation linked to this manufacturing methodology and can be clearly observed in the PLA13HA scaffolds (Figure 3c). The quantitative data of the HA particle distribution were obtained for the different 3D-printed PLA/HA scaffolds, registering the number of HA particles in relation to their position in distance from the surface. For it, and given that different parts of a single HA particle will be at different distances in depth from the cord surface, the average distance of each particle was the one calculated. These quantitative data are presented in Figure 7b for the PLA3HA scaffolds with infill from 60 to 100%. It can be first observed that the number of particles in each scaffold tends to increase with the percentage of infill, as expected, obtaining a maximum number of around 3000 particles in the case of PLA3HA-100%, with around 1000 particles for the same scaffold with the lowest infill. The frequency distribution of those particles was also obtained for all the scaffolds and presented in the inset of Figure 7b for the PLA3HA-80% scaffolds. At the micro level, related to the biocomposite cords, an exponential distribution of particles can be seen from the surface toward the interior of the cord, found in the first 80 μm (10% of the entire cord). This exponential distribution was repeated for the remaining samples in the series. This distribution would be beneficial for the application described by providing highly bioavailable HA and favoring the interaction with the cells without the need to wait for the entire degradation of the polymer.

Taking advantage of the binary 3D representation of the scaffolds obtained from the micro-CT processing of the images, with a resolution of 16 µm, calculations of porosity and connectivity (Euler characteristic) of the scaffolds were also performed. For it, a central cube or waffle of the scaffolds (with dimensions 7.27 × 2.37 mm) was defined to eliminate the edge effects and consider the volume of interest. In Figure 8, the volume of pores in relation to the total volume is presented for the PLA9HA scaffolds with infill from 60 to 100% (a) and the same relation for the scaffolds with 70% of infill varying the HA incorporation from 0 to 13 wt.% (b). It is first observed (Figure 8a) that the CT porosity linearly decreases with the infill percentage, as expected. These quantitative data revealed that the highest CT porosity in the volume of the PLA9HA scaffolds was obtained for the lowest infill, 60% and represented 76%, which was also the highest percentage of CT porosity obtained for all the scaffolds (provided as Appendix A). The lowest CT-porosity level quantified for these PLA9HA scaffolds was obtained for the highest infill, 100%, with a value of 39%. Therefore, the lowest infill promotes higher CT porosity and, according to the microscope images (Figure 2 and Figure 3), bigger pores. Taking into account all CT-porosity values, the lowest value was obtained for the PLA0HA-100% scaffolds, representing around 20% of the entire scaffold, with bulges and constrictions observed in the lines deposited leaving pores between adjacent printing filaments and layers. As previously mentioned, these PLA0HA-100% scaffolds presented square-shaped pores of around 440 µm from wall to front wall. In the case of the PLA9HA-100% scaffolds, the pore size was 300 µm. All of this seems to prove that the incorporation of HA gradually promotes the scaffolds to present higher CT porosity, in relation to the total volume of the scaffold, but with smaller pores. To confirm this influence of the HA incorporation into the scaffold porosity the CT porosity was also quantified and presented in Figure 8b for the PLA/HA scaffolds varying the HA incorporation from 0 to 13 wt.%. The results, presented for the infill of 70%, confirmed that CT porosity slightly increases with the content of HA in the printing mixture. However, this tendency breaks for 13% HA, where a decrease in the CT porosity is observed, together with a less clear linear decrease in the infill percentage (see Appendix A), indicating that the contribution of HA for this pellet 3D-printing methodology is limited to the amount values of 9 wt.% or lower. These results are in agreement with the study by Orozco et al. where they also establish a maximum of 10% by weight of HA for the manufacture of PLA/HA filaments since an increase in such concentration quickly becomes a problem for the processing and printing of the material, which prevents a uniform flow of the filament and results in imperfections and inconsistencies perceptible to the naked eye in the diameter of the filaments manufactured [24].

Finally, the connectivity of the pore network of the scaffolds was also quantified, as it is essential to facilitate the migration and proliferation processes of the cells, as well as the diffusion and exchange of nutrients and infiltration of the extracellular matrix [28]. Going into detail, the connectivity factor or Euler characteristic, N2(X), was the topological parameter obtained for all the scaffolds, and the results are presented in Table 4 and Figure 9. It was calculated from the skeleton of the pore network of the waffles using the Avizo Euler Number module, by the equation presented below consisting of the difference in the number of connected objects of one phase NC(X) (connected voids) minus the number of objects of the other phase separating them Nh(X) (printing cord):N2(X) = NC(X) − Nh(X)

Table 4 shows the N2(X) values found for all the 3D-printed PLA/HA scaffolds, where more negative values indicate a lesser number of connected paths. Thus, it can be observed that as the connectivity factor decreases, with higher negative values, the percentage of infill increases up to 90%. This decrease in connectivity is expected, as a higher number of printing cords reduces the number of interconnected pores (see the diagram in Figure 9). That is in good agreement with the CT porosity in the total volume (Figure 8, Appendix A), which decreases with increasing infill percentage. However, for 100% of infill, the connectivity values were equal to the ones at the lower infills, hence a high connectivity. This can be explained by the high density of the printing cords, which makes the software consider some paths of the pore network as closed, as it does not identify several printing cords and takes them as one (less infill). In addition, again, the PLA13HA scaffolds presented a more irregular behavior, in this case, of connectivity with the infill percentage. Moreover, these scaffolds (PLA13HA) presented the highest connectivity values for infill percentages of up to 90% in comparison to the scaffolds with a lower contribution of HA (PLA0HA, PLA3HA, PLA9HA) for the same corresponding infill. The aggregates and the high number of HA particles in these PLA13HA scaffolds promote it to present a high connectivity at a lower CT porosity in the total volume of the scaffold (according to the CT-porosity values).

The analysis of the particle size and distribution, together with the porosity and connectivity clearly revealed the influence of high amounts of HA on the pellet-based 3D-printing process, by modifying the melting process of PLA and affecting its rheological properties and, thus, the final print. Therefore, the need to use higher print flow rates implied a higher amount of material per minute coming out from the nozzle, resulting in a highly “fluid” material, which caused, in the case of the scaffolds with a contribution of 13 wt.% of HA, CT-porosity values lower than expected, together with a less clear linear decrease in CT porosity with the infill percentage and irregular behavior in terms of connectivity. On the other hand, the values of 39 to 76% in CT porosity and pore sizes in the range from 500 µm to 1 mm, depending on the infill percentage obtained for PLA9HA scaffolds are in agreement with the morphology of trabecular bone which consists of a porous environment with 50–90% porosity and pores up to 1 mm in diameter, surrounded by cortical bone [27,32] and with other works using different printing methodologies [1,42].

### 3.4. Biological Response In Vitro: Cell Proliferation

Finally, the biological response of the 3D-printed PLA/HA scaffolds was evaluated in vitro using the human osteosarcoma MG-63 cell line. Absorbance values obtained by using the MTS assay, directly proportional to cell viability, are presented in Figure 10 for the PLA0HA, PLA3HA, PLA9HA and PLA13HA scaffolds, all of them with 100% of infill. It can be observed that after 7 days of incubation, cell viability on the PLA3HA, PLA9HA and PLA13HA scaffolds with 100% of infill is confirmed to be at the same level (without statistically significant differences) as the scaffold without HA (PLA0HA) considered as the control. After 14 days of incubation, cell viability values were higher for all the scaffolds in comparison to the ones obtained after 7 days, which confirmed the proliferation of the cells seeded on the scaffolds. This cell proliferation rate presented a significantly higher value for the PLA13HA samples (*p* < 0.05). After 21 days, the cell viability registered for the scaffolds with HA in their composition was higher, in mean values, than the one obtained for the HA-free scaffold, with the PLA3HA scaffold being the one that significantly (*p* < 0.05) stands out from the others. Therefore, the cell proliferation data demonstrated that both the contribution of HA in the scaffolds of up to 13 wt.% and the porosity, pore size and described parameters associated with them for 100% of infill did not affect the cell proliferation and viability of MG63 cells. And, finally, the significant increase in the cell viability of all scaffolds from day 1 to day 21 confirmed the non-toxic properties of the 3D-printed PLA/HA scaffolds and their capability to promote cell proliferation.

Taking into consideration the two parameters evaluated, the influence of the HA contribution to the scaffolds (biochemical and mechanical) and the porosity (size, CT porosity in the total volume, connectivity), the cell proliferation results validated the 3D-printed PLA/HA scaffolds proposed. Going into detail, the HA contribution from 3 to 13 wt.% with the particles mostly distributed on the surface on each printing line (up to 30 µm in depth), providing higher HA bioavailability and surface roughness, was confirmed as beneficial for cell proliferation in vitro. The same occurred for the porosity, with square-shaped pores with sizes ranging from about 440 µm (PLA0HA) from wall to front wall to 380 µm in PLA3HA, 300 µm in PLA9HA and around 250 µm in PLA13HA, all of them with 100% of infill. The contribution of 3 wt.% of HA, with the value of about 36% porosity in volume, for the pores of 380 µm mentioned above of the PLA3HA-100% scaffolds, promoted the highest mean value (statistically significant (*p* < 0.05) from PLA0HA). Moreover, the contribution of 9 wt.% of HA, with the value of about 39% of porosity in volume, for pores of 300 µm of the PLA9HA-100% scaffolds, achieved the highest value of cell proliferation after up to 21 days. In both cases, most of the HA particles were not aggregated and were situated between 0 and 20 µm from the cord surface, ensuring higher biochemical availability and certain surface roughness. When reviewing the literature, enhanced bone formation and vascularization are reported for scaffolds with pore sizes larger than 300 µm [4,32], a size very similar to the scaffolds with 100% of the infill tested. These data are in agreement with other studies on PLA/HA printed scaffolds where other unconventional FDM 3D-printing processes were proposed, such as a mini-deposition system, producing PLA/HA scaffolds with a contribution of HA of 15 wt.%, porosities of 60% and pores sizes above 500 µm and proving the good proliferation of BMSCs [1]. Other authors proved the good biocompatibility of PLA scaffolds fabricated using conventional 3D printing (PLA/HA commercial filament) [24] or by the filament preparation by a melt-compounding using first an extruder [43]. The 3D-printing methodology proposed in the present work, based on direct 3D printing from a mixture of both materials, simplifies the processing and offers versatility, allowing the control of key parameters such as porosity and composition, given the behavior of both of them following a linear tendency in certain ranges. Moreover, the expected mechanical properties and the disposal of the HA particles close to the surface of each cord will provide the osteoblast with surface roughness and higher biochemical availability of the bioceramic, both beneficial for their functionality.

As mentioned earlier, previous studies have explored the combination of PLA and HA through 3D printing. However, these studies employed different methodologies compared to the approach used in the present work. Some studies involved the production of a composite filament beforehand [24], while others utilized premade pellets [1]. In contrast, this work focuses on directly mixing and printing the components. Through a systematic investigation that involved varying both the weight percentage of the HA incorporated and the infill percentage, we successfully validate the proposed methodology for manufacturing scaffolds with precise composition control and pore volume.

## 4. Conclusions

The feasibility of direct 3D printing from a mixture of the biomaterial polymer/ceramics, avoiding the need for pre-fabricated filaments, to obtain PLA/HA scaffolds, according to the most favorable requirements in terms of composition, porosity/connectivity and mechanical properties, is demonstrated. This work proves the versatility that this methodology offers with the control of the different scaffold parameters by adjusting the mixture and the infill percentages, to obtain modulated compositions and porous structures within the ranges reported in the literature to promote bone formation and vascularization. In fact, it opens interesting possibilities related to personalized medicine as its potential to directly load the scaffolds during the printing process with other biomolecules/drugs or nanomaterials of interest.

The details of the particular achievements provided in the present work are as follows:Three-dimensionally printed PLA/HA scaffolds with an increasing contribution of HA of up to 13 wt.% and uniformly distributed square-shaped pores, in the range from 250 µm to 850 µm, can be obtained with the proposed 3D-direct-printing methodology.This printing methodology creates scaffolds where HA particles are homogeneously distributed around the pores in the scaffold (macro level) and present an exponential distribution from the surface to the interior of the biocomposite cord (micro level) within the first 80 µm (10% of the entire cord diameter). This distribution favors the surface roughness and biochemical availability of HA.The fabrication methodology proposed guarantees a wide porosity range between 24 and 76% in total volume as a function of the percentage of infill applied and hydroxyapatite incorporated in the mixture.The mechanical properties of the scaffolds remain in expected values, with a trend of an increase in Young’s modulus with higher HA contribution.The expected in vitro cell viability and proliferation on the 3D-printed PLA/HA scaffolds obtained by this methodology were confirmed after up to 21 days.

## Figures and Tables

**Figure 1 polymers-15-02849-f001:**
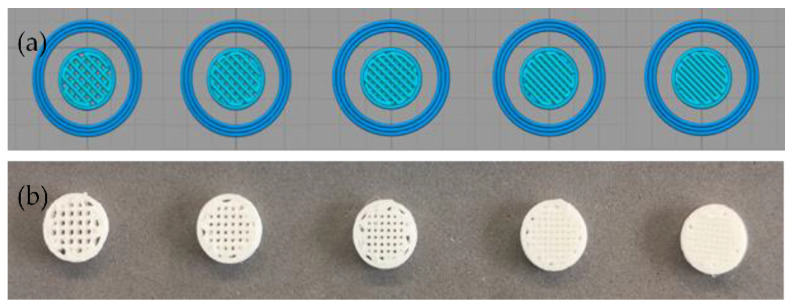
Simplify3D Professional Software images of the design from left to right from 60% to 100% of infill density (**a**) and images of the 3D-printed PLA9HA scaffolds obtained for one of the sets (**b**).

**Figure 2 polymers-15-02849-f002:**
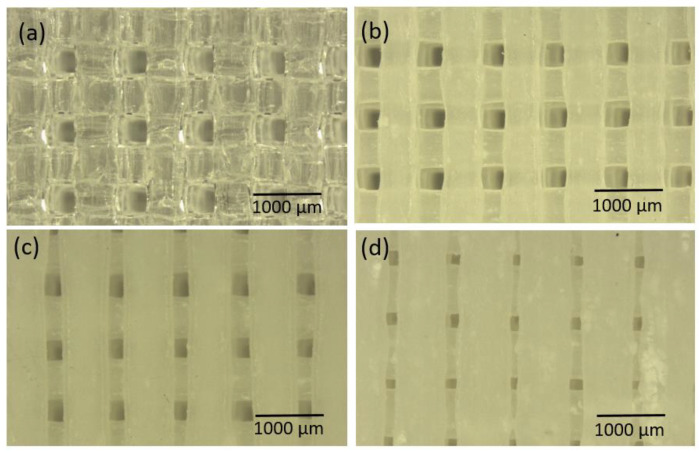
Magnified stereomicroscope images of PLA0HA scaffold with 100% infill (**a**), PLA3HA-100 (**b**), PLA9HA-100 (**c**) and PLA13HA-100 (**d**).

**Figure 3 polymers-15-02849-f003:**
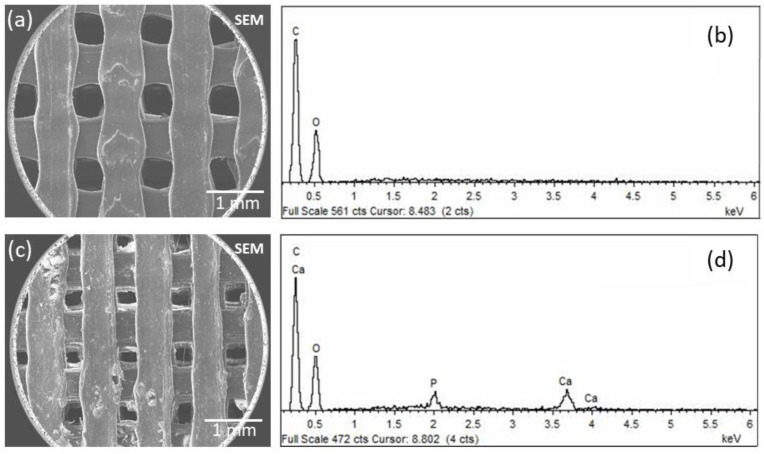
SEM micrograph and corresponding EDS of PLA0HA-80% infill (**a**,**b**) and PLA13HA-100% infill (**c**,**d**).

**Figure 4 polymers-15-02849-f004:**
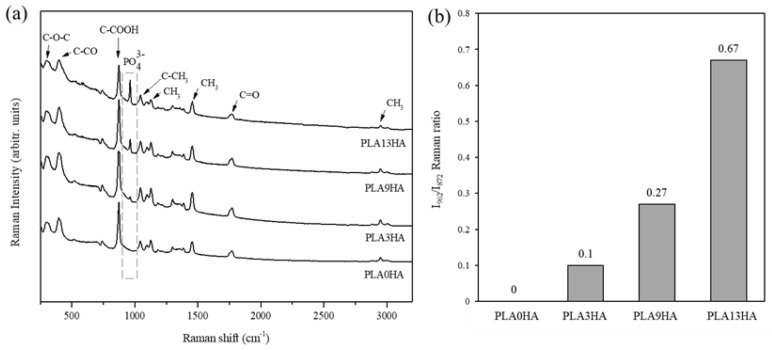
FT-Raman spectra of 3D-printed PLA and the PLA/HA scaffolds fabricated with 100% of infill. The main characteristic bands of 3D-printed PLA at the region of 250–3250 cm^−^^1^ are assigned, together with the characteristic main band of calcium phosphates (pointed square) (**a**). Quantitative evaluation of the FT-Raman spectra with the corresponding I962/I872 ratios for the 3D-printed PLA/HA scaffolds (**b**).

**Figure 5 polymers-15-02849-f005:**
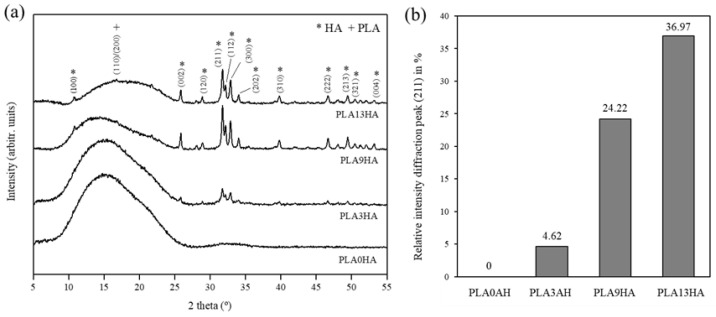
XRD diffraction patterns of PLA0HA and PLA/HA scaffolds in position 5–55° with crystal planes respectively attributed to PLA and HA are indicated (**a**). Quantitative evaluation of the diffraction peak intensity (211) in percentage as a function of wt.% HA incorporated into the scaffolds (**b**).

**Figure 6 polymers-15-02849-f006:**
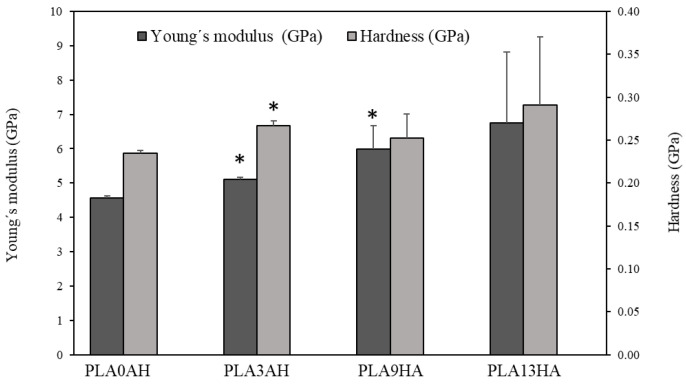
Young’s modulus and hardness of PLA discs with different concentrations of HA with 100% infill. The bar plot represents means ± standard errors. Statistical significance * (*p* ≤ 0.05) was determined with respect to the control, the scaffold without HA (PLA0HA).

**Figure 7 polymers-15-02849-f007:**
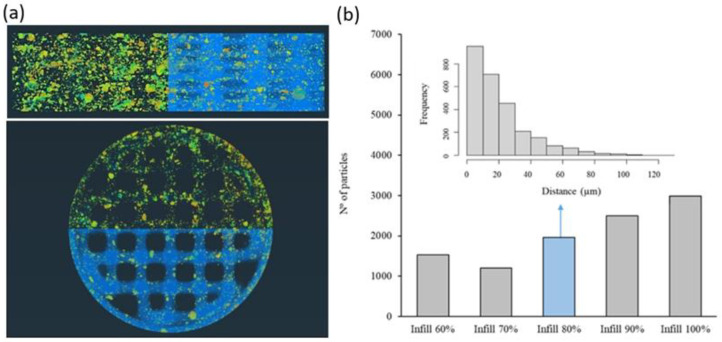
Micro-CT segmented images where HA particles (mainly orange, yellow and green colors) and the PLA matrix (blue color) of PLA13HA scaffold with 60% of infill (**a**) can be distinguished. Warmer colors of HA particles (orange) indicate proximity to the cord (printing filament) surface. Quantitative data in number of HA particles quantified for the PLA3HA scaffolds with 60 to 100% of infill (**b**) and inset of the frequency distribution of the HA particles in depth for the PLA3HA-80% scaffolds.

**Figure 8 polymers-15-02849-f008:**
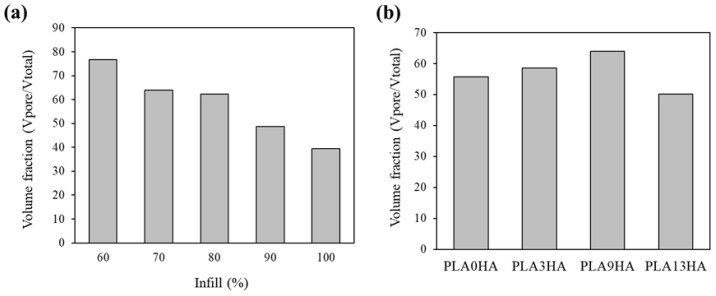
Volume fraction for PLA9HA scaffolds fabricated with different % of infill (**a**) and scaffolds with 70% of infill fabricated with different HA incorporations from 0 to 13 wt.% (**b**).

**Figure 9 polymers-15-02849-f009:**
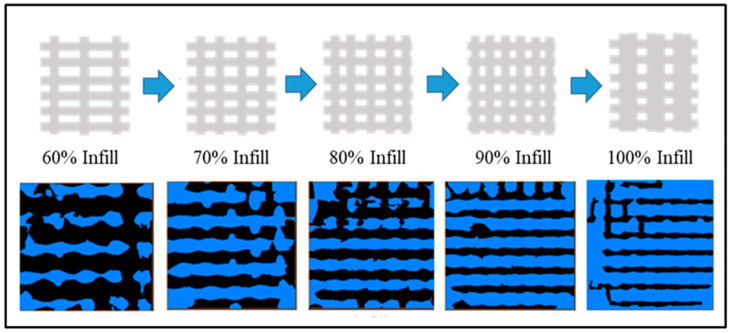
Illustrative diagram of the evolution of the connectivity factor.

**Figure 10 polymers-15-02849-f010:**
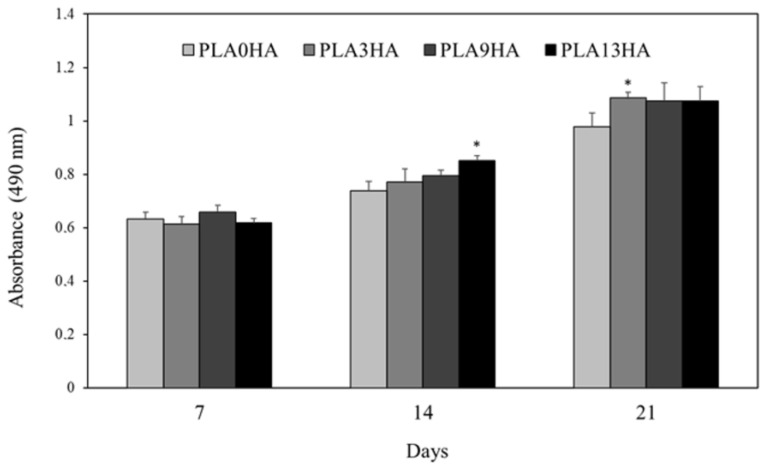
Cell proliferation (MTS assay) of the human osteosarcoma MG63 cell line after incubation for up to 7, 14 and 21 days on scaffolds of PLA0HA, PLA3HA, PLA9HA and PLA13HA. The bar chart represents mean ± standard error. Statistical significance was determined at * (*p* ≤ 0.05) with respect to the control, the scaffold without HA (PLA0HA).

**Table 1 polymers-15-02849-t001:** Technical data of SMARTFIL^®^ taken from the product data sheet. https://www.smartmaterials3d.com/pla-y-abs-pellets (accessed on 6 March 2023).

PLA Smartfil^®^	Value
Material density	1.24 g/cm^3^
Tensile strength	50 MPa
Tensile modulus	3.5 GPa
Flexural strength	83 MPa
Print temperature	200–240 °C
Glass transition temperature	60 °C

**Table 2 polymers-15-02849-t002:** Weight percentages (wt.%) of PLA and HA incorporated into each scaffold, as well as the percentage of infill applied and the corresponding labeling of the samples.

Scaffolds	Material (wt.%)		Infill (%)
	PLA	HA	
PLA0HA	100	0	60–100
PLA3HA	97	3	
PLA9HA	91	9	
PLA13HA	87	13	

**Table 3 polymers-15-02849-t003:** Main 3D printer parameters used.

Printer Tumaker NX Pro Pellets	Parameter Value, Range
Nozzle	0.8 mm
T_1_ (extruder 1)	140 °C
T_2_ (extruder 2)	220–240 °C
Bed temperature	45 °C
Infill density	60–100%
Infill pattern	Rectilinear (45/−45° angle)
Speed	1200 mm/min

**Table 4 polymers-15-02849-t004:** Connectivity factor (N2(X)) of the 3D-printed PLA/HA scaffolds.

Connectivity Factor	60% Infill	70% Infill	80% Infill	90% Infill	100% Infill
PLA0HA	−379	−331	−678	−698	−292
PLA3HA	−344	−378	−534	−709	−399
PLA9HA	−377	−432	−555	−676	−472
PLA13HA	−296	−342	−298	−599	−437

## Data Availability

Not applicable.

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
