# Peer review of "Physicochemical Properties of 3D-Printed Polylactic Acid/Hydroxyapatite Scaffolds"

_polymers, 2023, doi:10.3390/polym15132849_

Round 1

Reviewer 1 Report

The manuscript investigates the physicochemical properties of 3D printed PLA/HA scaffolds for bone tissue engineering applications. The comprehensive characterisation, including the HA distribution, pore size, mechanical properties, and in vitro evaluation, contributes valuable knowledge to the field. However, I suggest the authors consider addressing the following points for further improvement. Addressing these points will enhance the manuscript and contribute to its potential publication in Polymer journal.

1.      Provide more context and discuss the significance of the obtained results in relation to existing literature. This will help readers understand the novelty and advancement of the study in the field of bone tissue engineering.

2.      Consider discussing the limitations and potential challenges associated with the 3D printing technique used and the proposed PLA/HA scaffold system. Addressing these aspects will also provide a more balanced perspective and highlight areas for future research and development.

3.      Consider expanding the discussion on the in vitro evaluation, including additional analyses or results related to cell behavior, such as adhesion, migration, or differentiation. This would further strengthen the study's findings regarding the scaffolds' biocompatibility and potential for bone tissue engineering applications.

  • Use bullets to emphasise the main achievements of the paper

Reviewer 2 Report

Pérez-Davila et al. presented a strategy of using polylactic acid (PLA) and hydroxyapatite (HA) to construct 3D-printed scaffolds to tackle challenges in reconstructing or regenerating damaged bone tissues. The authors studied the physical and chemical properties of the 3D-printed scaffolds by varying the ratios between PLA and HA as well as the percentages of infills. This manuscript showed that HA was uniformly integrated into PLA matrix, promoting the properties of the composite in terms of Young’s moduli. Furthermore, in vitro tests demonstrated that bone cells could thrive on these scaffolds for up to 21 days, indicating their potential use in bone tissue regeneration. This is a comprehensive study with a list of characterization techniques performed to demonstrate the material’s properties.

 However, one essential point that authors did not clearly present is the major claim of this work. As the authors pointed out, integrating HA with PLA for 3D printing is not a new thing. Numerous studies have shown that PLA are combined with HA, either by surface modification or integrated as a single filament before being printed. Hence, what the authors are trying to bring to the community is a ‘one pot’ integration approach by modulating the PLA/HA ratio in the pellets form so that their composition can be varied. This is a brilliant selling point, but the authors did not show it is indeed a homogenous integration. Since both PLA and HA pellets are large particles, the blending process could be critical for the material properties, while this process was not mentioned and studied in the manuscript. Control experiment to compare with materials fabricated by PLA and HA as a single filament is also needed to highlight the advantage of the authors’ approach. Furthermore, the authors claimed that this is a homogenous integration, but the EDS experiment only showed the composition of the material at a specific point of acquisition. It would be necessary to show the mapping data to illustrate the uniformity of HA distribution in the polymer matrix. Therefore, this reviewer cannot recommend the publication of this manuscript before this major concern is addressed. Additionally, this reviewer has the following technical comments.

 1.      In Figure 2, all the optical microscopic images presented were from a fill ratio of 100% for (a) to (e) except for (f). It would be necessary to keep consistency here for the readers to follow the trend.

2.      In Figure 5, the authors employed PXRD to study the crystallinity of the 3D-printed scaffolds. This reviewer noticed that one diffraction peak form PLA was observed at a higher HA loading, but PLA is almost an amorphous phase in the absence of HA. Can the authors explain that?

3.      In Figure 6, the authors investigated the mechanical properties of the 3D-printed scaffolds. From this result, a conclusion of improving the stiffness of PLA with the integration of HA can be barely drawn, given the high standard deviation for PLA13HA. P values should be calculated using ANOVA with post-hoc Tukey tests to show the statistical significance. It would be suggested that the authors look into the uniformity of the materials to see if this can account for the high standard deviation.

4.      The manuscript is constructed with long sentences, and it is hard for readers to follow, particularly in the introduction section. It would be much clearer if the authors could split them into several short sentences.

The manuscript is constructed with long sentences, and it is hard for readers to follow, particularly in the introduction section. It would be much clearer if the authors could split them into several short sentences.

Round 2

Reviewer 2 Report

The authors have addressed this reviewer's comments. This reviewer would like to recommend the publication of this manuscript.